# Architecture Matters: MetaFormer and Global-aware Convolution Streaming for Image Restoration

## Abstract

Transformer-based methods have sparked significant interest in this field, primarily due to their self-attention mechanism's capacity to capture long-range dependencies. However, existing transformer-based image restoration methods restrict self-attention on windows or across channels to avoid computational complexity explosion, limiting their ability to capture long-range dependencies. This leads us to explore the following question: Is the general architecture abstracted from Transformers significantly impact the performance of existing Transformer-based image restoration methods? To this end, we first analyze the existing attention modules and replace them with solely convolution modules, also known as *convolution streaming*. We demonstrate that these convolution modules deliver comparable performance with existing attention modules at the similar cost of computation burden. Our findings underscore the importance of the overall Transformer architecture in image restoration, motivating the principle of *MetaFormer*-a general architecture abstracted from transformer-based methods without specifying the feature mixing manners. To further enhance the capture of long-range dependencies within the powerful MetaFormer architecture, we construct an efficient global-aware convolution streaming module with Fourier Transform. Integrating the MetaFormer architecture and global-aware convolution streaming module, we achieves consistent performance gain on multiple image restoration tasks including image deblurring, image denoising, and image deraining, with even less computation burden.

## 1 Introduction

Image restoration aims to recover high-quality images from their low-quality counterparts by removing degradations (*e.g.*, blur, noise), laying the foundation for different vision tasks. Since image restoration is a highly ill-posed problem, model-based image restoration methods are usually derived from physical principles or statistical assumptions, e.g., priors. Due to the strong ability to learn image priors from large-scale datasets, Convolutional Neural Networks (CNNs) emerge as a successful alternative for image restoration.

Recently, transformer models have achieved remarkable success in NLP tasks Vaswani et al. (2017) and high-level vision tasks Carion et al. (2020). One typical feature of the above transformer models is the self-attention mechanism, which shows a strong ability to capture long-range dependencies. Since 2021, the breakthroughs from transformer networks have sparked great interest in image restoration. However, the quadratic increase in computational complexity and memory consumption with image size has limited the direct application of self-attention to image restoration, especially for modern high-resolution images. Inspired by window-based self-attention Liu et al. (2021), the mainstream remedy is to apply self-attention in local windows Liang et al. (2021); Wang et al. (2022); Chen et al. (2022b); Xiao et al. (2022). For instance, SwinIR Liang et al. (2021) is among the first to adopt window-based self-attention in image restoration. Restormer Zamir et al. (2022) applies self-attention across channel dimension instead of spatial dimension.

Although the above transformer-based methods have achieved significant performance gain, restricting self-attention to local windows or across channels fails to fully utilize self-attention for depen-

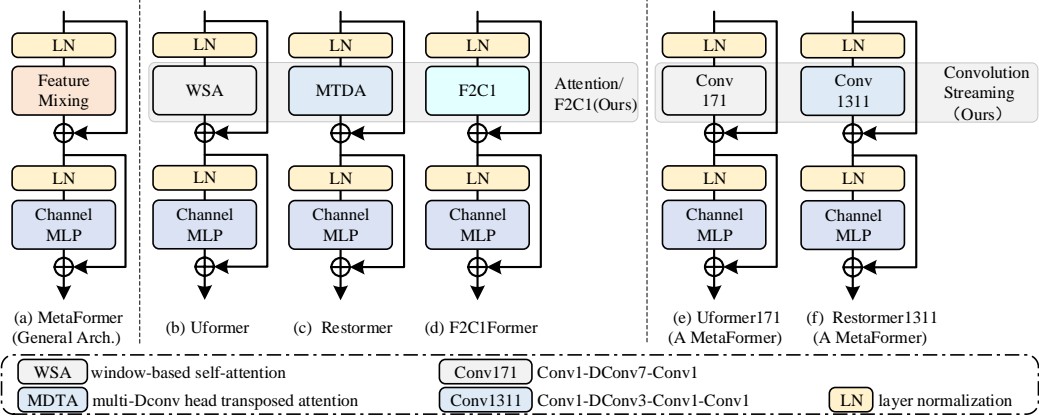

Figure 1: Illustration of representative transformer blocks and MetaFormer. (a) MetaFormer (proposed general architecture), (b) the transformer block in Uformer Wang et al. (2022), (c) the transformer block in Restormer Zamir et al. (2022), (d) our proposed block in our F2C1Former, (e) our proposed block in Uformer171, (f) our proposed block in Restormer1311.

dencies capturing of pixels in the long range. Based on this consideration, this paper aims to answer the following question: *whether the general architecture in Fig. 1a, matters the performance of existing transformer-based methods?*

To answer the question, our paper's journey begins with a comprehensive analysis of existing attention modules, replacing them with a stack of convolution modules, referred to as convolution streaming.

Our findings demonstrate that these convolution modules achieve comparable performance to attention modules under the similar complexity, as shown in Fig. 2. Therefore, we posit that the general transformer architecture matters the promising performance levels observed. This realization propels us to introduce the principle of MetaFormer, a general image restoration architecture. As shown in Fig. 1a, the basic architecture of MetaFormer is LN + FeatureMixing + LN + ChannelMLP. Here, LN stands for Layer Normalization, and examples of FeatureMixing as well as Channel MLP are shown in Fig. 3 and Fig. 4.

Although convolution streaming equivalents have demonstrated the effectiveness, capturing global dependencies is still very important for image restoration, since many image degradation processes share global statistic. To further enhance the capture of long-range dependencies within the powerful MetaFormer architecture, we construct an efficient global-aware convolution streaming module with Fourier Transform, named FourierC1C1 (F2C1). Our F2C1 is simple and easy to implement by performing Fourier transform and inverse Fourier transform on both ends of convolution modules. In other words, convolution operations are performed in the Fourier domain. Integrating the MetaFormer architecture and global-aware convolution streaming module, we can significantly improve the performance of image restoration tasks with less computation burden.

The main contributions are summarized as follows:

- MetaFormer Principle: We propose the principle of MetaFormer, a general architecture abstracted from transformer-based methods without specifying the feature mixing manners. Experimental results demonstrate that MetaFormer matters the performance of existing transformer models.

- Convolution Streaming Equivalents: We conduct an in-depth analysis of the existing attention mechanisms from the mathematical models. Correspondingly, we also propose simplified convolution streaming counterparts for each representative attention.

- Global-aware Convolution Streaming: Within the powerful MetaFormer architecture, we construct the global-aware convolution streaming module with Fourier Transform. Integrating the MetaFormer architecture and global-aware convolution streaming module, we

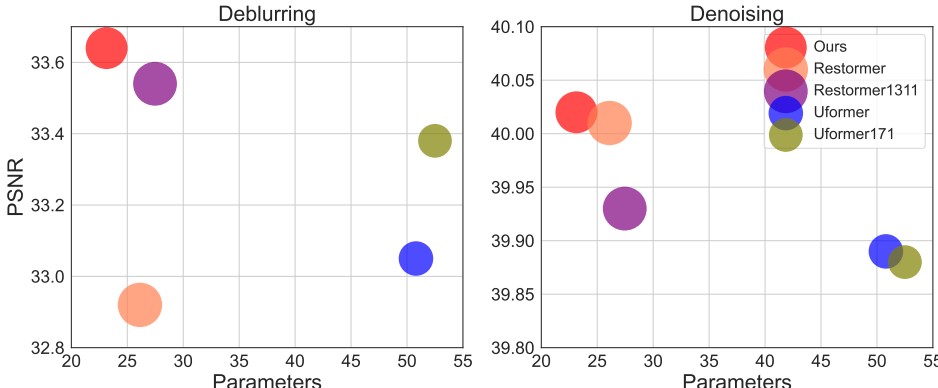

Figure 2: Model Performance *vs.* Parameters *vs.* MACs. The area of the circles indicate the relative value of MACs. By replacing the attention modules with simplified convolution streaming versions, Uformer171/Restormer1311 achieves significant performance gain for deblurring (GoPro) and comparable results for denoising (SIDD) over Uformer/Restormer.

> achieve promising performance on nine datasets of multiple image restoration tasks including image deblurring, denoising, and deraining with less computation burden.

The proposal of MetaFormer does not diminish the importance of self-attention, but offers a broader architectural perspective in image restoration. It can still instantiate FeatureMixing as self-attention, but not limited to self-attention. Our work calls for more future research to explore more effective architectures. Additionally, the derived models, *i.e.*, Uformer171 and Restormer1311, are expected to serve as simple baselines for future image restoration research.

## 2 RELATED WORK

### 2.1 IMAGE RESTORATION

Conventional model-based methods focus on designing image priors (*e.g.*, total variation prior Chan & Wong (1998), channel prior Yan et al. (2017), gradient prior Chen et al. (2019); Pan et al. (2017)) to constrain the solution space for effective image restoration. Data-driven CNNs have been shown to surpass conventional model-based methods since they can learn more generalizable image priors from large-scale data. Among the CNN-based methods, several widely embraced methodologies include the encoder-decoder-based U-Net, skip connections, and spatial/channel attention. The U-Net, introduced in Ronneberger et al. (2015), has been extensively verified its effectiveness in image restoration due to its multi-scale processing mechanisms Tao et al. (2018); Zamir et al. (2021); Cho et al. (2021); Cui et al. (2023b); Tu et al. (2022). Furthermore, skip connections He et al. (2016) have proven suitable for image restoration since the degradations in images can be seen as residual signals Zamir et al. (2021); Zhang et al. (2017b; 2019b; 2021); Cui et al. (2023b). Besides, spatial/channel attention is commonly incorporated since they can selectively strengthen useful information and inhibit useless information Zamir et al. (2021); Li et al. (2018); Zhang et al. (2018b); Suin et al. (2020) from the spatial/channel dimension. *Despite the remarkable success of CNN-based restoration methods over the past half-decade, they encounter challenges in modelling long-range dependencies, which are critical for effective image restoration.*

### 2.2 VISION TRANSFORMERS

The first transformer model is proposed for translation tasks Vaswani et al. (2017), and has rapidly achieved remarkable success in different NLP tasks. Motivated by the success in NLP, many researchers have applied transformers to high-level vision tasks Touvron et al. (2021); Kolesnikov et al. (2021). Notably, ViT Kolesnikov et al. (2021) learns the mutual relationships of a sequence of patches cropped from an image. The typical feature of the above vision transformers is the self-attention mechanism which has the strong ability to capture long-range dependencies.

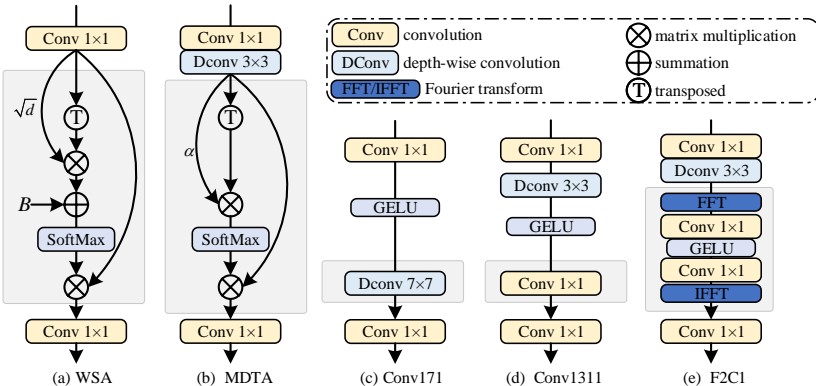

Figure 3: Representative attention modules and simplified convolution streaming modules for mixing features (FeatureMixing). (a) WSA in Uformer Wang et al. (2022), (b) MDTA in Restormer Zamir et al. (2022), (c) Conv171 in Uformer171, (d) Conv1311 in Restormer1311, (e) Proposed F2C1.

Since 2021, the breakthroughs from transformer in high-level vision tasks have sparked great interest in low-level vision tasks such as super-resolution Liang et al. (2021); Chen et al. (2022b); Li et al. (2023), denoising Wang et al. (2022); Chen et al. (2021a); Xiao et al. (2022), and deblurring Zamir et al. (2022); Tsai et al. (2022). However, due to the quadratic increase in complexity and memory consumption with respect to the number of pixels, applying self-attention to high-resolution images—a common requirement in image restoration—becomes infeasible. To tackle this challenge, the mainstream is to employ self-attention at the patch/window level Chen et al. (2021a); Liang et al. (2021); Wang et al. (2022) or channel Zamir et al. (2022) level. However, it comes at the expense of capturing global dependencies.

Recent endeavors by Xiao et al. (2023); Li et al. (2023); Zhou et al. (2023) have merged to focus on global modelling in image restoration. For instance, GRL Li et al. (2023) proposes anchored stripe attention for global modelling, albeit with a complexity increase from $\mathcal{O}(H^2)$ to $\mathcal{O}(H^3)$, where $H$ is the height and weight of the square input. ShuffleFormer Xiao et al. (2023) presents a random shuffle strategy to model non-local interactions with local window transformer. The strategy extends the local scope without introducing extra parameters, but need compute the attention map multiple times, and thus consumes more resources (running time or memory). Fourmer, as devised by Zhou et al. (2023), customizes Fourier spatial interaction modelling and Fourier channel evolution for image restoration, featuring a core advantage of being lightweight and striking a favorable balance between parameters and performance.

To sum up, most existing transformer-based methods mainly focus on how to efficiently calculate self-attention. *In contrast to these methods, we encapsulate existing transformer-based techniques within the MetaFormer framework, examining them from a general framework perspective.* For global modelling, current works either need extra computation resource (memory or computation cost) or prefer to lightweight models. *Different from these methods, we propose a global-aware convolution streaming F2C1, capturing the global dependency and enjoying the powerful MetaFormer architecture.In fact, integrating F2C1 and MetaFormer not only achieves the state-of-the-art performance but also further demonstrates the effectiveness of the generalized architecture MetaFormer.*

## 3 METHODOLOGY

### 3.1 MOTIVATION

Recently, transformer-based methods have achieved promising performance in image restoration, where the self-attention mechanism capturing long-range dependencies is considered to be one of the key reasons for its success. To reduce the computation and memory burden of self-attention, IPT Chen et al. (2021a) computes self-attention on patches of size $48 \times 48$ cropped from an image. A line of methods applies self-attention on local windows, *i.e.*, window-based attention Liu et al. (2021), such as SwinIR Liang et al. (2021) and Uformer Wang et al. (2022). The above methods

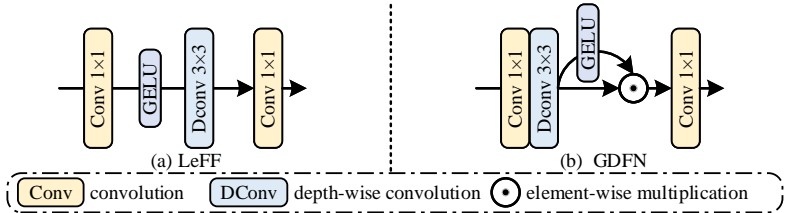

Figure 4: Illustration of Channel MLP modules in representative transformer-based image restoration methods. (a) Locally-enhanced feed-forward network (LeFF) in Uformer Wang et al. (2022), (b) Gated-Dconv feed-forward network (GDFN) in Restormer Zamir et al. (2022).

have not fully exploited global dependencies. To cope with the issues while remaining efficient, Restormer Zamir et al. (2022) applies self-attention across channel dimension instead of spatial dimension. Although the above transformer-based methods have achieved significant performance gain over CNN-based models, *existing attention modules focus on efficiently compute self-attention. However, it struggles for capturing long-range modelling capabilities, defying the main motivation of using self-attention.*.

The above analysis motivates us to examine a fundamental question: *Does the general architecture of transformers matter the advanced performance of current transformer-based techniques?* Our conclusion is that the general architecture of transformers, MetaFormer, matters. In the rest of this paper, we keep other factors such as U-Net configurations and training strategies unchanged, for fair comparison.

## 3.2 METAFORMER

We begin by introducing the concept MetaFormer. As shown in Fig. 1a, the attention module is replaced with the FeatureMixing module while the other components are kept the same as conventional transformers. We denote that the input features $X$ of a MetaFormer block are of size $C \times H \times W$, where $C$, $H$, and $W$ denote the number of channels, the height, and the weight, respectively.

In particular, the MetaFormer block consists of two sub-blocks. The first sub-block can be mathematically expressed as

$$Y = X + \text{FeatureMixing}\left(\text{LN}(X)\right), \tag{1}$$

where $Y$ is the output of the first sub-block. LN denotes layer normalization Ba et al. (2016). FeatureMixing represents a module for mixing features. We plot some examples of FeatureMixing in existing transformer-based methods in Fig. 3.

The second sub-block is expressed as

$$O = Y + \text{ChannelMLP}\left(\text{LN}(Y)\right), \tag{2}$$

where ChannelMLP represents the module for non-linear transformation, which consists of channel expansion and reduction operations. Some examples of ChannelMLP in existing transformer-based methods can be found in Fig. 4.

**Instantiations of MetaFormer** *By specifying the designs of FeatureMixing and ChannelMLP in MetaFormer, different transformer blocks can be obtained. If FeatureMixing and ChannelMLP are specified as window-based attention (WSA), and locally-enhanced feed-forward network (LeFF), respectively, MetaFormer degenerates into Uformer Wang et al. (2022). If FeatureMixing and ChannelMLP are specified as multi-Dconv head transposed attention (MDTA), and Gated-Dconv feed-forward network (GDFN), respectively, MetaFormerFormer degenerates into Restormer Zamir et al. (2022). It is worth noting that MeatFormer do not deny the role of self-attention in image restoration. MetaFormer includes FeatureMixing, which can be instantiated as self-attention.*

## 3.3 EXISTING ATTENTIONS AND ITS CONVOLUTION VERSIONS

As demonstrated in Section 3.1, current transformer-based methods mainly focus on conducting self-attention efficiently. In contrast, we pay attention to the general architecture, MetaFormer.

In this paper, we argue that the general architecture MetaFormer matters the success of transformer-based methods. To verify this, our solution is conducting experiments by replacing attention modules with solely convolution modules, and comparing the performance. Without loss of generality, we consider two representative attention modules, *i.e.*, window-based self-attention (WSA) in Uformer Wang et al. (2022), and multi-Dconv head transposed attention (MDTA) in Restormer Zamir et al. (2022). Through our analysis, we propose two stacks of convolutions (convolution streaming), *i.e.*, Conv171 and Conv1311, as simplified approximation. Taking the convolution streaming as the FeatureMixing, we instantiate MetaFormer in a convolution streaming manner as Uformer171 and Restormer1311, respectively. In particular, we have plotted the illustration of the attention mechanisms and the convolution streaming approximation in Fig. 3, and the MetaFormer instantiations in Fig. 1.

### 3.3.1 UFORMER171

Uformer applies self-attention in local regions using swin transformer design Liu et al. (2021), *i.e.*, window-based self-attention (WSA), as shown in Fig. 3a. The process following Eq. 1 is

$$
\begin{aligned}
Y &= X + W_{1\times1}\text{Attention}\left(Q, K, V\right), \\
\text{Attention}\left(Q, K, V\right) &= \text{SoftMax}(QK^T/\sqrt{d} + B)V, \\
[Q, K, V] &= W_{1\times1}\text{LN}(X),
\end{aligned} \tag{3}
$$

where $Q, K, V \in \mathbb{R}^{M^2 \times d}$ are the query, key, and value. $M$ is the window size. $d$ is the dimension of the query/key. $W_{1\times1}$ represents the $1 \times 1$ convolution. $B \in \mathbb{R}^{M^2 \times M^2}$ is the relative position bias. Since WSA realizes spatial information interaction in the local window of size $8 \times 8$ ($M = 8$), the computational complexity is reduced from quadratic to linear with respect to the number of pixels. However, this manner also restricts its capabilities for long-range dependency modelling Chu et al. (2022), although with shifted window approach. Based on this, we simplify the attention in WSA to a $7 \times 7$ depth-wise convolution. It is worth noting that we do not aim to get the exact equivalent form of WSA, but an approximate convolution-equivalent form. Correspondingly, we simplify WSA as Conv171 (Conv1-Dconv7-Conv1), and term the model as Uformer171 (Fig. 1e).

### 3.3.2 RESTORMER1311

Restormer Zamir et al. (2022) applies self-attention across channel dimension instead of spatial dimension, and proposes multi-Dconv head transposed attention (MDTA) in Fig. 3b. The process following Eq. 1 is

$$
\begin{aligned}
Y &= X + W_{1\times1}\text{Attention}\left(Q, K, V\right), \\
\text{Attention}\left(Q, K, V\right) &= \text{SoftMax}(QK^T/\alpha)V, \\
[Q, K, V] &= W_{3\times3}^d W_{1\times1}\text{LN}(X),
\end{aligned} \tag{4}
$$

where $Q, K, V \in \mathbb{R}^{C \times HW}$. $\alpha$ is a learnable scale factor. $W_{3\times3}^d$ represents the $3 \times 3$ depth-wise convolution. MDTA in essence performs a linear transformation on $V$ in the channel dimension. Although the computation complexity is linear with respect to the number of pixels, it aggregates pixel-wise information across channels, and thus we simplify the attention in MDTA into a $1 \times 1$ convolution. Correspondingly, we simplify MDTA as Conv1311 (Conv1-Dconv3-Conv1-Conv1), and term the model as Restormer1311 (Fig. 1f).

### 3.4 F2C1FORMER

Current work mainly focus on efficiently computing self-attention, but at the expense of capturing global dependencies. Although several solutions have been proposed for global modelling, these methods either introduce extra computation cost Li et al. (2023); Xiao et al. (2023) or compromise the performance Zhou et al. (2023). Different from these methods, we propose a global-aware convolution streaming F2C1 based on the effectiveness of convolution streaming equivalents and the global property brought by Fourier transform. In the next, we first introduce Fourier transform. Then, we elaborate on the proposed F2C1.

Table 1: Computation cost, parameters, and memory comparison.

| Module | Computation Cost | Parameters | Memory |
|---|---|---|---|
| WSA Wang et al. (2022) | $4HWC^2 + 2HWCs^2$ | $4C^2$ | $8HWC + hHWs^2$ |
| MDTA Zamir et al. (2022) | $4HWC^2 + 2HWC^2/h$ | $4C^2$ | $5HWC + C^2/h$ |
| F2C1 (Ours) | $2HWC^2 + 2HWC^2/h$ | $2C^2 + 2C^2/h$ | $5HWC$ |

### 3.4.1 PRELIMINARY

Fourier transform is a widely used signal processing and analysis tool. For an image or a feature with multiple channels, the Fourier transform is applied to each channel separately. Given a 2D signal $x \in \mathbb{R}^{H \times W}$, the Fourier transform $\mathscr{F}$ turns it to Fourier domain as $\mathscr{F}(x)$

$$\mathscr{F}(x)(u,v) = \frac{1}{\sqrt{HW}} \sum_{h=0}^{H-1} \sum_{w=0}^{W-1} x(h,w) e^{-j2\pi\left(\frac{h}{H}u + \frac{w}{W}v\right)} \tag{5}$$

where $(u,v)$ are the coordinates in Fourier domain. We use $\mathscr{F}^{-1}(x)$ to defines the inverse Fourier transform. For the frequency representation $\mathscr{F}^{-1}(x)$, there are two utilizable properties: 1) According to Eq. 5, arbitrary pixel at $(u,v)$ is involved with all the pixels in the original domain (image or feature). In other words, $\mathscr{F}^{-1}(x)$ is a inherently global representation, which enjoys elegant theoretical guarantees in global modelling. 2) Both $\mathscr{F}(x)$ and $\mathscr{F}^{-1}(x)$ can be efficiently implemented with FFT and iFFT, respectively.

### 3.4.2 PROPOSED MODULE F2C1

By performing Fourier transform and inverse Fourier transform on both ends of convolution modules, our F2C1, shown in Fig 3(e), is formulated as

$$\begin{aligned} Y &= X + W_{1\times1}\text{Global}(X_e), \\ X_e &= W_{3\times3}^d W_{1\times1}\text{LN}(X), \end{aligned} \tag{6}$$

The core component $\text{Global}(X_e)$ consists of three steps: 1) Fourier transform, 2) feature transformation, and 3) inverse Fourier transform. Specifically, given features $X_e$, we first apply Fourier transform to $X$ to obtain $\mathscr{F}^{-1}(X_e)$. Then we conduct feature transformation using an MLP (two $1 \times 1$ convolutions with GELU in between). Finally, we transfer the obtained features back to the original domain with inverse Fourier transform. Overall, the $\text{Global}(X_e)$ is formulated as

$$\text{Global}(X_e) = \mathscr{F}^{-1}\left(W_{1\times1}^2 \sigma W_{1\times1}^1(\mathscr{F}(X_e))\right), \tag{7}$$

where $\sigma$ represent the GELU non-linearity. Following multi-head self-attention, we divide channels into different heads, and learn the interactions in each head parallelly. This design also reduces the parameters and computation cost.

We also list the computation cost, parameters, and Memory of our F2C1 in Table 1. For comparison, we also include those of MSA and MDTA. The results demonstrate that Compared with WSA and MDTA, F2C1 is more compute- and storage-friendly.

## 4 EXPERIMENTS

### 4.1 SETUP

We conduct extensive experiments on nine datasets including image deblurring, image denoising, and image detraining. For image deblurring, GoPro Nah et al. (2017), a widely used dataset, is adopted. For image denoising, we adopt the widely used SIDD Abdelhamed et al. (2018) dataset. For the effectiveness of F2C1Former, we conduct extra experiments on image deraining task. Rain14000 Fu et al. (2017b), Rain1800 Yang et al. (2017), Rain800 Zhang et al. (2020a), Rain100H Yang et al. (2017), Rain100L Yang et al. (2017), Rain1200 Zhang & Patel (2018), and Rain12 Li et al. (2016) are adopted.

Table 2: Quantitative results of Uformer Wang et al. (2022) and Restormer Zamir et al. (2022), and corresponding convolution streaming versions: Uformer171, and Restormer1311 on GoPro and SIDD.

| | Deblurring (GoPro) | | Denoising (SIDD) | | | |
|---|---|---|---|---|---|---|
| | PSNR | SSIM | PSNR | SSIM | Para. (M) | Macs. (G) |
| Uformer | 33.05 | 0.962 | 39.89 | 0.960 | 50.80 | 85.47 |
| Uformer171 | 33.38 | 0.965 | 39.88 | 0.960 | 52.50 | 81.57 |
| Restormer | 32.92 | 0.961 | 40.01 | 0.960 | 26.13 | 140.99 |
| Restormer1311 | 33.54 | 0.965 | 39.93 | 0.960 | 27.48 | 137.51 |

Following Zamir et al. (2022); Wang et al. (2022); Chen et al. (2022a), we adopt PSNR and SSIM Wang et al. (2004) as the evaluation metrics for quantitative experiments. The implementation details are given in the supplementary materials.

We assess both MetaFormer and F1C2Former to address the following questions:

- The Impact of Architecture: Does the architectural choice significantly influence results? Can the generalized MetaFormer, incorporating straightforward convolution streaming, attain state-of-the-art performance? (Section 4.2)
- Enhancing Image Restoration: Does the inclusion of F1C2Former, an augmentation to MetaFormer featuring non-attention-based global modeling, lead to improved image restoration performance. (Section 4.3)

## 4.2 EFFECTIVENESS OF METAFORMER

### 4.2.1 MOTION DEBLURRING

We give the quantitative results on GoPro in Table 2. The visual results are given in the supplementary materials.

**Uformer171 *vs*. Uformer Wang et al. (2022).** By replacing WSA with Conv171 (Conv1-Dconv7-Conv1), Ufomer171 achieves competitive results with Uformer on GoPro. Specifically, Uformer171 achieves 0.33 dB performance gain on GoPro over Uformer.

**Restormer1311 *vs*. Restormer Zamir et al. (2022).** By replacing MDTA with Conv1311 (Conv1-Dconv3-Conv1-Conv1), Restormer1311 outperforms Restormer by 0.62 dB in terms of PSNR on GoPro.

### 4.2.2 REAL IMAGE DENOISING

We give the quantitative results on SIDD in Table 2. The visual results are given in the supplementary materials.

**Uformer171 *vs*. Uformer.** By replacing WSA with Conv171 (Conv1-Dconv7-Conv1), Ufomer171 achieves competitive results with Uformer on SIDD with comparable complexity and parameters. Specifically, Uformer171 brings 0.01 dB PSNR loss on SIDD over Uformer.

**Restormer1311 *vs*. Restormer.** By replacing MDTA with Conv1311 (Conv1-Dconv3-Conv1-Conv1), Restormer1311 achieves 0.08 dB PSNR loss over Restormer on SIDD with comparable complexity and parameters.

The above results demonstrate that the general architecture, MetaFormer, matters the performance of transformer-based image restoration methods.

## 4.3 MORE COMPARISONS WITH RECENT ADVANCES

### 4.3.1 MOTION DEBLURRING

Table 3 gives the quantitative results on the GoPro dataset. F2C1Former (Ours) delivers the state-of-the-art performance. Compared with ShuffleFormer which aims to achieve non-local interactions, F2C1Former brings 0.28 dB PSNR improvement. The visual results are given in the supplementary materials.

Table 3: Quantitative results on the GoPro dataset (single image motion deblurring).

| Methods | PSNR | SSIM |
|---|---|---|
| DeblurGAN Kupyn et al. (2018) | 28.70 | 0.858 |
| Nah et al. Nah et al. (2017) | 29.08 | 0.914 |
| Zhang et al. Zhang et al. (2018a) | 29.19 | 0.931 |
| DeblurGAN-v2 Kupyn et al. (2019) | 29.55 | 0.934 |
| SRN Tao et al. (2018) | 30.26 | 0.934 |
| Gao et al. Gao et al. (2019) | 30.90 | 0.935 |
| DBGAN Zhang et al. (2020b) | 31.10 | 0.942 |
| MT-RNN Park et al. (2020) | 31.15 | 0.945 |
| DMPHN Zhang et al. (2019a) | 31.20 | 0.940 |
| Suin et al. Suin et al. (2020) | 31.85 | 0.948 |
| SPAIR Purohit et al. (2021) | 32.06 | 0.953 |
| MIMO-UNet+ Cho et al. (2021) | 32.45 | 0.957 |
| IPT Chen et al. (2021a) | 32.52 | - |
| MPRNet Zamir et al. (2021) | 32.66 | 0.959 |
| HINet Chen et al. (2021b) | 32.71 | 0.959 |
| Uformer Wang et al. (2022) | 33.06 | **0.967** |
| Restormer Zamir et al. (2022) | 32.92 | 0.961 |
| MAXIM-3S Tu et al. (2022) | 32.86 | 0.961 |
| Stripformer Tsai et al. (2022) | 33.08 | 0.962 |
| Stoformer Xiao et al. (2022) | 33.24 | 0.964 |
| SFNet Cui et al. (2023b) | 33.27 | 0.963 |
| ShuffleFormer Xiao et al. (2023) | 33.38 | 0.965 |
| IRNeXt Cui et al. (2023a) | 33.16 | 0.962 |
| **F2C1Former (Ours)** | **33.64** | 0.966 |

Table 4: Quantitative results on the SIDD dataset (real image denoising).

| Methods | PSNR | SSIM |
|---|---|---|
| DnCNN Zhang et al. (2017a) | 23.66 | 0.583 |
| BM3D Dabov et al. (2007) | 25.65 | 0.685 |
| CBDNet Guo et al. (2019) | 30.78 | 0.801 |
| RIDNet Anwar & Barnes (2019) | 38.71 | 0.951 |
| AINDNet Kim et al. (2020) | 38.95 | 0.952 |
| VDN Yue et al. (2019) | 39.28 | 0.956 |
| SADNet Chang et al. (2020) | 39.46 | 0.957 |
| DANet+ Yue et al. (2020) | 39.47 | 0.957 |
| CycleISP Zamir et al. (2020a) | 39.52 | 0.957 |
| MIRNet Zamir et al. (2020b) | 39.72 | 0.959 |
| DeamNet Ren et al. (2021) | 39.35 | 0.955 |
| MPRNet Zamir et al. (2021) | 39.71 | 0.958 |
| HINet Chen et al. (2021b) | 39.99 | 0.958 |
| NBNet Cheng et al. (2021) | 39.75 | 0.959 |
| DAGL Mou et al. (2021) | 38.94 | 0.953 |
| Uformer Wang et al. (2022) | 39.89 | **0.960** |
| Restormer Zamir et al. (2022) | 40.02 | **0.960** |
| MAXIM-3S Tu et al. (2022) | 39.96 | **0.960** |
| CAT Chen et al. (2022b) | 40.01 | **0.960** |
| ShuffleFormer Xiao et al. (2023) | 40.00 | **0.960** |
| **F2C1Former (Ours)** | **40.02** | **0.960** |

Table 5: Quantitative results on the Rain14000 dataset (image deraining).

| Method | DerainNet Fu et al. (2017a) | SEMI Wei et al. (2019) | DIDMDN Zhang & Patel (2018) | UMRL Yasarla & Patel (2019) | RESCAN Li et al. (2018) | PreNet Ren et al. (2019) | MSPFN Jiang et al. (2020) |
|---|---|---|---|---|---|---|---|
| PSNR | 24.31 | 24.43 | 28.13 | 29.97 | 31.29 | 31.75 | 32.82 |
| SSIM | 0.861 | 0.782 | 0.867 | 0.905 | 0.904 | 0.916 | 0.930 |
| Method | MPRNet Zamir et al. (2021) | HINet Chen et al. (2021b) | SPAIR Purohit et al. (2021) | Restormer Zamir et al. (2022) | MAXIM-2S Tu et al. (2022) | SFNet Cui et al. (2023b) | **F2C1Former** Ours |
| PSNR | 33.64 | 33.91 | 33.34 | **34.18** | 33.80 | 33.69 | **34.18** |
| SSIM | 0.938 | 0.941 | 0.936 | 0.944 | 0.943 | 0.937 | **0.945** |

### 4.3.2 REAL IMAGE DENOISING

Table 4 gives the quantitative results on the SIDD dataset. F2C1Former (Ours) achieves competitive results. F2C1Former achieves the highest PSNR. Compared with Restormer, F2C1Former has fewer parameters and less computation burden, as illustrated in Table 1. The visual results are given in the supplementary materials.

### 4.3.3 IMAGE DERAINING

Table 5 gives the quantitative results on the Rain14000 dataset. F2C1Former (Ours) perform favourably against other methods. Compared with recent method SFNet, F2C1Former achieves 0.49 dB PSNR improvement. The visual results are given in the supplementary materials.

## 5 CONCLUSIONS

Within this paper, we abstracted the attention modules in existing transformer-based methods, and proposed a general image restoration structure termed MetaFormer, which matters the performance of existing transformer-based models. To enhance the of capture long-range dependencies, we also propose a global-aware convolution streaming F2C1. By specifying the feature mixing module as F2C1, the integrated F2C1Former achieves superior results on multiple image restoration tasks including image deblurring, denoising, and deraining.

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
