The supplementary materials are organized as follows:

- Section A gives the implementation details including network configurations and training strategies.
- Section B presents more visual results.
- Section C discusses the limitations of our work.
- Section D gives the broader impacts of our work.

# A   IMPLEMENTATION DETAILS

We elaborate on the implementation details in terms of network configurations and training strategies. For convenience, we summarize the implementation details in Table 6. *It is worth noting that since Uformer and Restormerare equipped with different network configurations and training strategies, we train Uformer171, and Restormer1311 with their respective settings.* For better illustration, we give the U-Net architectures in Figure 5. All the experiments are conducted on NVIDIA-SMI A100 GPUs.

*Network Configurations.* The input projection and output projection are implemented as $3 \times 3$ convolutions, as shown in Figure 5.

For **Uformer171**, we adopt a 4-level U-Net. The channel dimension is set to 32, and the channel expansion factor in Channel MLP is set to 4 as those of Uformer Wang et al. (2022). DownSample is implemented using a $4 \times 4$ convolution with stride 2, and UpSample is implemented using a $2 \times 2$ transposed convolution with stride 2. For the depths of encoder, bottleneck, and decoder, Uformer is set to [1, 2, 8, 8], 2, and [8, 8, 2, 1], respectively. To keep the comparable parameters and complexity, we adjust the depths of encoder, bottleneck, and decoder of Uformer171 to [1, 2, 8, 10], 3, and [10, 8, 2, 1], respectively.

For **Restormer1311**, we adopt a 3-level U-Net. The channel dimension is set to 48, and the channel expansion factor in Channel MLP is set to 2.66 as those of Restormer Zamir et al. (2022). DownSample is implemented using a $3 \times 3$ convolution with stride 1 and the PixelUnshuffle operation, and UpSample is implemented using

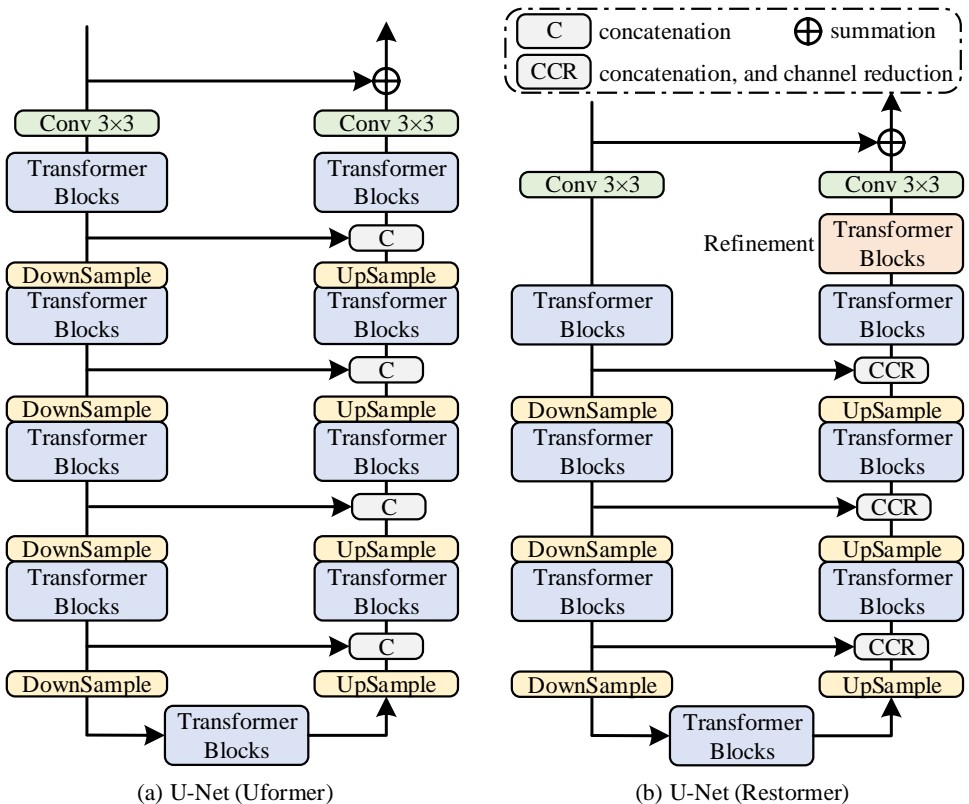

(a) U-Net (Uformer)                    (b) U-Net (Restormer)

Figure 5: U-Net architectures for representative transformer-based image restoration methods.   (a) Uformer Wang et al. (2022), (b) Restormer Zamir et al. (2022).

Table 6: Details of network configurations and training strategies for different models.

| | | Uformer Wang et al. (2022) | Uformer171 | Restormer Zamir et al. (2022) | Restormer1311 |
|---|---|---|---|---|---|
| Network configurations | Input Projection | convolution (k=3, s=1) | convolution (k=3, s=1) | convolution (k=3, s=1) | convolution (k=3, s=1) |
| | Output Projection | convolution (k=3, s=1) | convolution (k=3, s=1) | convolution (k=3, s=1) | convolution (k=3, s=1) |
| | Levels of U-Net | 4 | 4 | 3 | 3 |
| | Depths of Encoder | [1, 2, 8, 8] | [1, 2, 8, 9] | [4, 6, 6] | [4, 6, 8] |
| | Depths of Bottleneck | 2 | 3 | 8 | 9 |
| | Depths of Decoder | [8, 8, 2, 1] | [9, 8, 2, 1] | [6, 6, 4] | [8, 6, 4] |
| | Channel Dimension | 32 | 32 | 48 | 48 |
| | DownSample | convolution (k=4, s=2) | convolution (k=4, s=2) | convolution (k=3, s=1), PixelUnshuffle | convolution (k=3, s=1), PixelUnshuffle |
| | UpSample | transposed convolution (k=2, s=2) | transposed convolution (k=2, s=2) | convolution (k=3, s=1), PixelShuffle | convolution (k=3, s=1), PixelShuffle |
| | Channel Expansion Factor in Channel MLP | 4 | 4 | 2.66 | 2.66 |
| Training strategies | Training Patch Size | 256×256 (deblurring), 128×128 (denoising) | 256×256 (deblurring), 128×128 (denoising) | 128, 160, 192, 256, 320, 384 (Progressive) | 128, 160, 192, 256, 320, 384 (Progressive) |
| | Batch Size | 8 (deblurring), 32 (denoising) | 8 (deblurring), 32 (denoising) | 64, 40, 32, 16, 8, 8 (Progressive) | 64, 40, 32, 16, 8, 8 (Progressive) |
| | Epochs | 3000 (deblurring), 250 (denoising) | 3000 (deblurring), 250 (denoising) | | |
| | Iterations | | | 300K | 300K |
| | Optimizer | AdamW ((0.9, 0.999), 0.02) | AdamW ((0.9, 0.999), 0.02) | AdamW ((0.9, 0.999), $10^{-4}$) | AdamW ((0.9, 0.999), $10^{-4}$) |
| | Initial Learning Rate | $2 \times 10^{-4}$ | $2 \times 10^{-4}$ | $3 \times 10^{-4}$ | $3 \times 10^{-4}$ |
| | Final Learning Rate | $1 \times 10^{-6}$ | $1 \times 10^{-6}$ | $1 \times 10^{-6}$ | $1 \times 10^{-7}$ |
| | Decay Strategy | Cosine | Cosine | Cosine | Cosine |
| | Data Augmentation | horizontal flipping or rotation (90°, 180°, or 270°) | horizontal flipping or rotation (90°, 180°, or 270°) | horizontal or vertical flipping | horizontal or vertical flipping |
| | Loss functions | Charbonnier loss | Charbonnier loss | L1 loss | L1 loss |

a $3 \times 3$ convolution with stride 1 and the PixelShuffle operation. For the depths of encoder, bottleneck, and decoder, Restormer is set to [4, 6, 6], 8, and [6, 6, 4], respectively. To keep the comparable parameters and complexity, we adjust the depths of encoder, bottleneck, and decoder of Restormer1311 to [4, 6, 8], 9, and [8, 6, 4], respectively. In the refinement stage, we keep the number of blocks to 4 as those of Restormer.

*Training Strategies.* For fair comparison, we train Uformer171, Restormer1311, and Baseline131/NAFNet131 with the same train strategies as those of Uformer Wang et al. (2022), Restormer Zamir et al. (2022), and Baseline/NAFNet Chen et al. (2022a), respectively.

Specifically, for **Uformer171**, we randomly crop patches of size $256 \times 256$ from the original training image pairs for training. We train Uformer171 with a batch size of 8 for deblurring and 32 for denoising by 3K epochs. The AdamW with weight decay 0.02 and momentum terms (0.9, 0.999) is used for optimization. The learning rate declines from $2 \times 10^{-4}$ to $1 \times 10^{-6}$ with the cosine decay strategy Loshchilov & Hutter (2016). The training data is randomly augmented with horizontal flipping or rotation (90°, 180°, or 270°). For Uformer171, we use the Charbonnier loss for training as those of Uformer Wang et al. (2022).

For **Restormer1311**, we adopt the progressive learning strategy as those of Restormer. Specifically, we crop patches of size $512 \times 512$ with step size 256 from the training data. We train Restormer1311 for 300K iterations. The AdamW with weight decay $10^{-4}$ and momentum terms (0.9, 0.999) is used for optimization. We first train Restormer1311 with the patch size $128 \times 128$, batch size 64, and learning rate $3 \times 10^{-4}$ for 92K. Then, the patch size and batch size pairs are updated to $[(160^2, 40), (192^2, 32), (256^2, 16), (320^2, 8), (384^2, 8)]$ at iterations [92K, 156K, 204K, 240K, 276K], and the learning rate correspondingly declines from $3 \times 10^{-4}$ to $1 \times 10^{-6}$ with the cosine decay strategy. The training data is randomly augmented with horizontal or vertical flipping. For Restormer1311, we use the L1 loss for training as those of Restormer Zamir et al. (2022).

# B MORE VISUAL RESULTS

Figure 6 presents more visual results on GoPro Nah et al. (2017). Figure 7 presents a visual example on SIDD Abdelhamed et al. (2018). Figure 8 presents a visual example on GoPro Nah et al. (2017). Figure 9 presents a visual example on SIDD Abdelhamed et al. (2018). Figure 10 presents a visual example on Rain14000 Fu et al. (2017b).

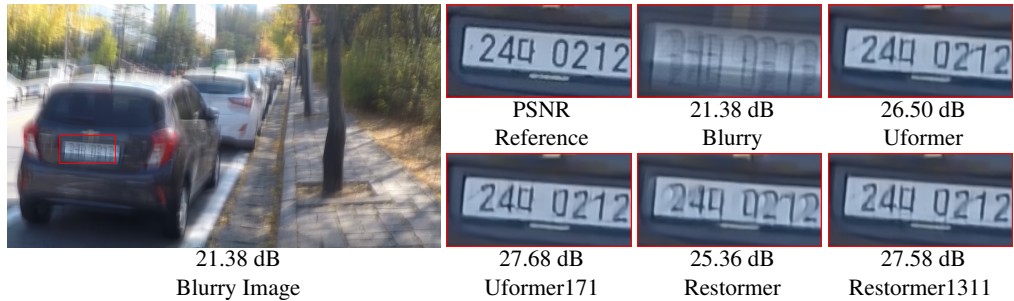

Figure 6: **Single image motion deblurring** on the GoPro dataset Nah et al. (2017). Compared to Uformer Wang et al. (2022), and Restormer Zamir et al. (2022), Uformer171 and Restormer1311 generate sharper and visually more faithful results.

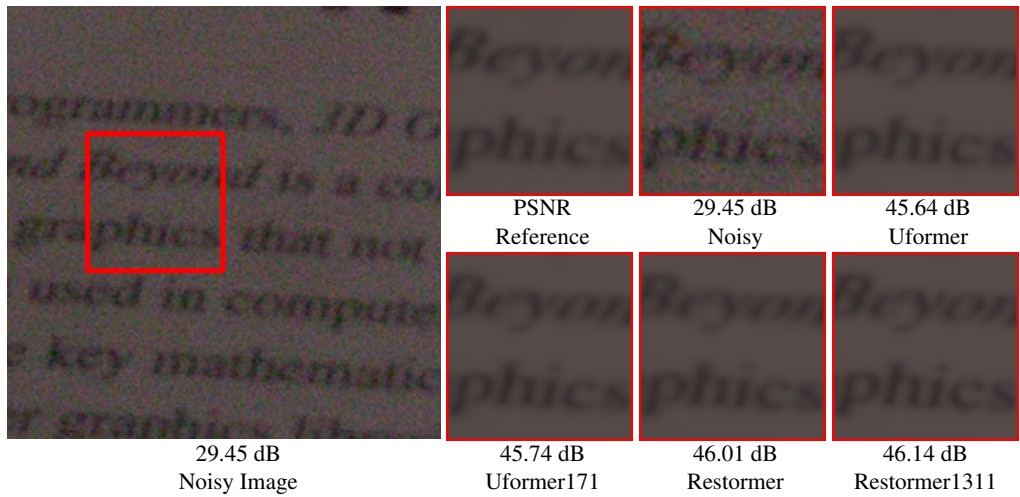

Figure 7: **Real image denoising** on the SIDD dataset Abdelhamed et al. (2018). Compared to Uformer Wang et al. (2022), and Restormer Zamir et al. (2022), Uformer171 and Restormer1311 generate cleaner and visually more faithful results.

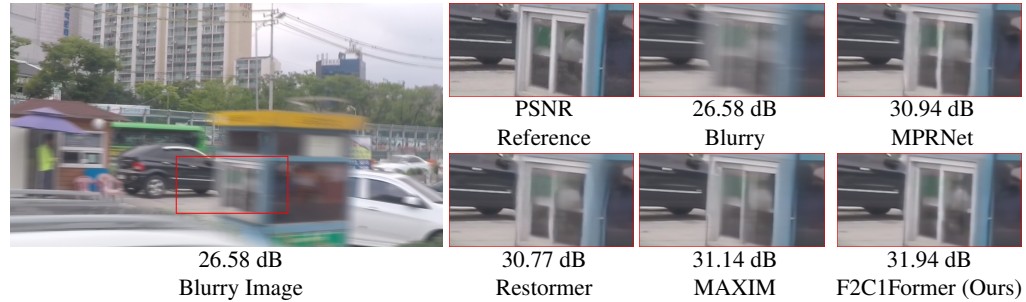

Figure 8: **Single image motion deblurring** on the GoPro dataset Nah et al. (2017). Compared to MPRNet Zamir et al. (2021), Restormer Zamir et al. (2022), and MAXIM Tu et al. (2022), F2C1Former(Ours) generates sharper and visually more faithful results.

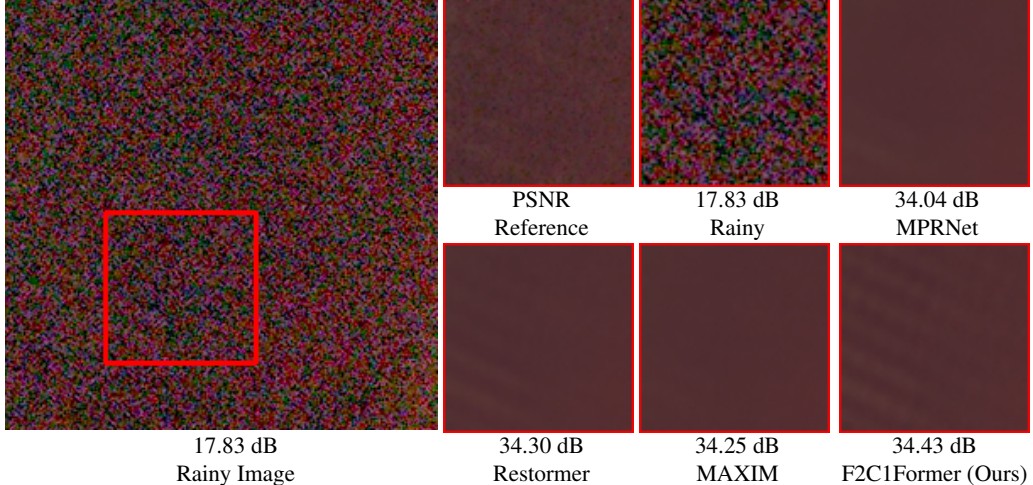

Figure 9: **Real image denoising** on the SIDD dataset Abdelhamed et al. (2018). Compared to MPRNet Zamir et al. (2021), Restormer Zamir et al. (2022) and MAXIM Tu et al. (2022), F2C1Former(Ours) generates cleaner and visually more faithful results.

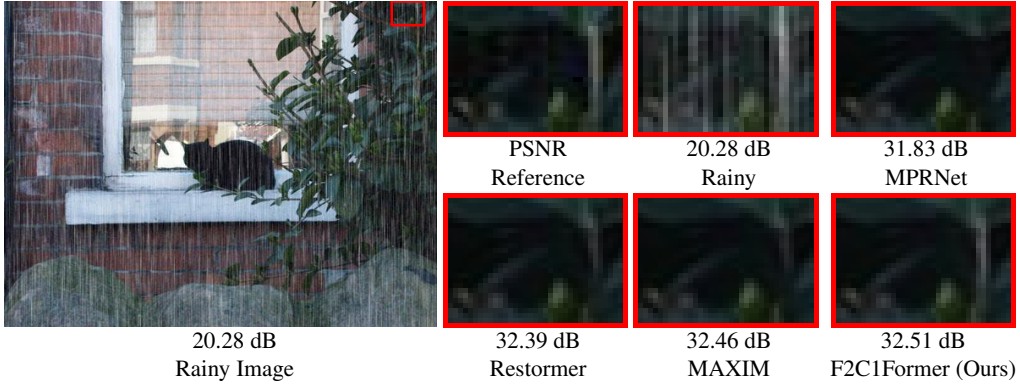

Figure 10: **Single image deraining** on the Rain14000 dataset Fu et al. (2017b). Compared to MPRNet Zamir et al. (2021), Restormer Zamir et al. (2022), and MAXIM Tu et al. (2022), F2C1Former(Ours) generates sharper and visually more faithful results.

## C  LIMITATIONS AND DISCUSSIONS

We have demonstrated the effectiveness of F2C1Former on image restoration tasks including image deblurring, image denoising and deraining. More experiments could be conducted on other image restoration tasks, such as dehazing, desnowing, *etc*.

Notwithstanding the extensive and promising results, there are still open questions remaining for transformer-based image restoration methods.

**Is there a more effective transformer-like structure for image restoration?** The success of MetaFormer demonstrates that the structure LN + FeatureMixing + LN + ChannelMLP matters for image restoration, even without attention modules. This reminds us that an effective structure is also vital for image restoration. Since MetaFormer is abstracted from transformers proposed first for NLP tasks, exploring structures customized for image restoration is also a meaningful task.

# D BROADER IMPACTS

Nowadays, image acquisition systems inevitably suffer from various degradations due to camera shakes, bad weather conditions, *etc*. Image restoration solves the problem well, and has important research and application value. Our proposed MetaFormer, such as Uformer171, Restormer131, and F2C1Former, achieve competitive restoration results. However, there are also potential negative societal impacts. For example, image restoration technology is prone to privacy leakage.