# OpenReview forum: "ARCHITECTURE MATTERS: METAFORMER AND GLOBAL-AWARE CONVOLUTION STREAMING FOR IMAGE RESTORATION"
_ICLR.cc/2024/Conference — ICLR 2024 Conference Withdrawn Submission_

### Official Review · Reviewer_Dq6b · 2023-10-31

**Soundness:** 3 good
**Presentation:** 3 good
**Contribution:** 2 fair
**Rating:** 3
**Confidence:** 5

**Summary:**

This paper proposes a MetaFormer principle that the general architecture of transformers-based methods, without specific feature mixing manners, is important for image restoration. The authors conduct experiments on motion deblurring and real image denoising to show that Uformer and Restormer with convolution streaming modules can achieve comparable or better performance, compared to using attention modules. A new image restoration network F2C1 is constructed by using the MetaFormer architecture and a global-aware streaming module with Fourier Transform. Experimental results show that F2C1 obtains state-of-the-art performance on image motion deblurring, real image denoising and image deraining.

**Strengths:**

1. The authors provide detailed analysis to illustrate the commonality of existing transformer-based image restoration networks and show the importance of the MetaFormer architecture for network design.
2. The design of F2CIFormer is sound and the experimental results on motion deblurring and deraining look good.
3. The overall writing is well and the paper is easy to read.

**Weaknesses:**

1. The paper provides limited new insights. Using convolution streaming modules (Or even MLP modules) in the meta transformer block (i.e., MetaFormer in this paper) can build the comparable network with attention-based network is not a new knowledge in the computer vision community. Many existing works have shown similar conclusions[1][2][3][4][5].
2. The conducted experiments mainly focus on motion deblurring and denoising. I think this is a tricky presentation, because these two tasks (or rather Gopro and SIDD datasets) are relatively simple and there have been works proving that methods without powerful attention can achieve very high performance [4][5]. On the contrary, existing works[6][7] demonstrate that attention mechanism, especially window-based self-attention is of great significance to the task of image super-resoluion, while this paper avoids this task.
3. The proposed method F2C1Former integrates the fourier transform to construct the network, but there is no experimental analysis to show the impact of this design. This design is not novel either, as it appears on many existing works [8][9][10].

Overall, I think this paper has limited technical contributions and cannot bring new inspiration.

[1] A ConvNet for the 2020s. CVPR 2022.
[2] MLP-Mixer: An all-MLP Architecture for Vision. NeurIPS 2021.
[3] MetaFormer Is Actually What You Need for Vision. CVPR 2022.
[4] Simple Baselines for Image Restoration. ECCV 2022.
[5] MAXIM: Multi-Axis MLP for Image Processing. CVPR 2022.
[6] SwinIR: Image Restoration Using Swin Transformer. ICCVW 2021.
[7] Activating More Pixels in Image Super-Resolution Transformer. CVPR 2023.
[8] Fast Fourier Convolution. NeurIPS 2020.
[9] Resolution-robust Large Mask Inpainting with Fourier Convolutions. WACV 2022.
[10] SwinFIR: Revisiting the SwinIR with Fast Fourier Convolution and Improved Training for Image Super-Resolution. Arxiv 2022.

**Questions:**

How does the computational complexity compare between F2C1Former and several state-of-the-art methods in Tab.3, 4, 5?

---

### Official Review · Reviewer_EQ2P · 2023-10-31

**Soundness:** 2 fair
**Presentation:** 1 poor
**Contribution:** 1 poor
**Rating:** 1
**Confidence:** 5

**Summary:**

This paper mainly proposes MetaFormer, which generalizes the several plain self-attention networks. While some transformer-based image restoration methods employ unique attention mechanisms or feed forward networks (FFN), the authors change them into Feature Mixing layers and channel MLP, respectively. Their Feature Mixing replaces the self-attention layers of Uformer and Restormer with convolutional layers. Moreover, they also propose F2C1Former, employing Fast Fourier Transform and convolution operations in Feature Mixing to capture global context. With the proposed components, MetaFormer and F2C1Former show comparable results on several image restoration tasks.

**Strengths:**

[S1] The proposed Uformer171 and Restormer1311 slightly decreased the computational complexity. These architectures achieved better performance than their baseline models on a deblurring task.

[S2] F2C1Former highly outperformed state-of-the-art deblurring methods.

**Weaknesses:**

[W1] There are many unnecessary contents, such as Fig.4 and Sec.3.4.1, which can be replaced with other explanations or figures. More important contents should be included to improve this paper. The essential experiments or visualizations are absent.

[W2] MetaFormer architecture is too general because some alternatives to LN-Attn-LN-FFN architecture have been proposed and proven to be effective in this domain. For example, the architecture of Swin V2 [1], Attn-LN-FFN-LN, outperformed that of MetaFormer [2, 3]. Moreover, while ELAN [4] discarded conventional FFN (substituted by 1x1 shift convs and placed before Attn), it can show outstanding performance. However, the authors do not conduct the experiments that compare the plain LN-Attn-LN-FFN structure with other effective architectures.

[W3] The proposed F2C1 does not have novelty. Many existing image restoration methods tried to apply Frequency-based approaches to their networks, such as Fourier Transform or Wavelet Transform. Nevertheless, F2C1 simply places FFT and IFFT operators before and after 1x1 conv, which is not different from existing works. In fact, if some (novel) visual or theoretical impacts of utilizing frequency domain were provided to present how FFT and IFFT can make the network capture global context, F2C1’s novelty can be re-considered. But there are not any relevant demonstrations or ablation studies, thus, Sec.3.4 fails to justify both motivation and effectiveness of the F2C1 component.

[W4] It is not clear how effective 7x7 depth-wise conv of Uformer and 1x1 conv of Restormer is. While both Uformer171 and Restormer1311 get better performances than baselines on a deblurring task, they do not seem careful choices. Some ablation studies of structure variants can be considered. Or, it is required to thoroughly analyze how 7x7 dconv and 1x1 conv can act as window self-attention of Uformer and channel self-attention of Restormer, respectively.

[W5] What makes the above replacements (W4) improve the networks is not presented. While this paper emphasizes the significance of capturing global dependency, it appears inconsistent with the main idea that alternative convolution, which is generally considered to lack global dependency, can outperform self-attention. Did you visualize intermediate feature maps?

[W6] The illustrated visual comparisons cannot show obviously improved performance (Figs.6-9 of Appendix). Even if presented PSNR score is higher than others or baselines, the results of the proposed methods are not visually pleasant. More careful cherry-picking is demanded.

Ref)

[1] Liu, Ze, et al. "Swin transformer v2: Scaling up capacity and resolution." Proceedings of the IEEE/CVF conference on computer vision and pattern recognition. 2022.

[2] Conde, Marcos V., et al. "Swin2SR: Swinv2 transformer for compressed image super-resolution and restoration." European Conference on Computer Vision. Cham: Springer Nature Switzerland, 2022.

[3] Choi, Haram, Jeongmin Lee, and Jihoon Yang. "N-gram in swin transformers for efficient lightweight image super-resolution." Proceedings of the IEEE/CVF Conference on Computer Vision and Pattern Recognition. 2023.

[4] Zhang, Xindong, et al. "Efficient long-range attention network for image super-resolution." European Conference on Computer Vision. Cham: Springer Nature Switzerland, 2022.

**Questions:**

Please see weaknesses.

How about narrowing the domain from general image restoration to deblurring task since the improvements on that task were notable. The reviewer thinks that focusing on significantly improving a specific task and comprehensively analyzing the impacts of proposed components makes this paper more compelling and insightful. I sincerely hope my major concerns will be properly addressed to improve your paper, and the later updated version will be accepted by next conference or journal.

---

### Official Review · Reviewer_7HFZ · 2023-10-31

**Soundness:** 3 good
**Presentation:** 3 good
**Contribution:** 2 fair
**Rating:** 3
**Confidence:** 5

**Summary:**

The paper shows that the overall architecture of Transformer matters the performance of existing transformer models by substituting the elements of Transformer with convolutional layers. Furthermore, the paper proposes a global-aware convolution streaming module based on the Fourier Transform. The network obtains promising performance on three image restoration tasks.

**Strengths:**

The paper is easy to follow. The paper reveals that the overall architecture of Transformer matters the performance of existing transformer models. The proposed F2C1 module introduces the global modeling ability into MetaFormer. The experiments span three tasks.

**Weaknesses:**

1. The novelty of the paper is limited. Firstly, the role of the overall architecture of Transformer has been extensively discussed in high-level vision tasks. The authors only conduct experiments on two tasks, i.e., deblurring and denoising. However, even with these two tasks, the conclusion is not unanimous. For example, the convolution version increases the performance of deblurring while decreasing that of denosing. Secondly, the Fourier transform technique is widely used in low-level vision tasks, especially the form of using 1x1 convolutions between FFT and IFFT.
2. The experiments are not convicing enough because the authors only select one dataset for each task.

**Questions:**

1. The reviewer doubts the logic of introducing the problem. What led the authors to explore the metaformer? The motivation is farfetched.
2. The authors expect Uformer171 and Restormer1311 to serve as simple baselines for image restoration. However, there is no comparison between these models with the related baseline model, such as NAFNet.
3. Uformer performs self-attention within local windows and the reviewer thinks it still has quadratic complexity with respect to pixels.
4. What do s and h in Tab.1 stand for?
5. The authors divide channels into different heads in F2C1. Does this operation decrease the performance?
6. The improvement of F2C1 is not discussed separately. By comparing Tab.2 and Tab.3/4, the reviewer thinks that the improvement is minimal.
7. The reviewer suggests that the authors provide the performance on HIDE and the other four deraining datasets used in Restormer and SFNet. Following these algorithms, the authors can directly apply the pre-trained models to these datasets without additional training.

---

### Official Review · Reviewer_98jJ · 2023-11-01

**Soundness:** 1 poor
**Presentation:** 2 fair
**Contribution:** 1 poor
**Rating:** 3
**Confidence:** 5

**Summary:**

This paper tries to answer the question of whether the the general architecture of transformer is important for image restoration. To answer this question, the authors start by analyzing the attention module in previous works such as Uformer and Restormer. Based on that, the authors proposed convolutional versions of the transformers. Experiments are done on a couple of image restoration tasks including single-image motion deblurring, image deraining, and image denoising on SIDD dataset.

**Strengths:**

1. The paper is well-written and easy to follow.

2. Apart from the writing, the motivation is also quite good.

**Weaknesses:**

1. Since this paper aims high and intends to solve the general architecture problem, it is not only important to describe the details of the introduced blocks (e.g. Conv171 and Conv1311) but also important to explain the motivation and rationale of those designs.

2. This paper claimed a couple of modules such as Conv171, Conv1311, and Fourier transform. But no ablation study is done. This would make the readers question the effectiveness of the proposed modules.

3. The computational burden of the proposed F2C1 is not necessarily low. Compared with a vanilla transformer with three convolutions, 5 convolutions are used in F2C1, let alone the additional operation of FFT and IFFT. This might not be an efficient solution.

4. State-of-the-art method on single-image motion deblurring is not compared [1]. The authors mentioned computational complexity in Sec. 3.4. A fair comparison with those methods would include quantitative results, computational complexity, and the number of parameters.

5. Misleading results and claim. The authors claimed that for image deraining, experiments are done on Rain14000 Fu et al. (2017b), Rain1800 Yang et al. (2017), Rain800 Zhang et al. (2020a), Rain100H Yang et al. (2017), Rain100L Yang et al. (2017), Rain1200 Zhang & Patel (2018), and Rain12 Li et al. (2016). But after searching the main paper and the supplementary results, only the experimental results on Rain14000 dataset are reported. It is well-known that the different deraining datasets have quite diverse characteristics. State-of-the-art performance on one test set does not mean good performance on all of the datasets.

6. The importance of the introduced Fourier transform is not ablated.

[1] Efficient and Explicit Modelling of Image Hierarchies for Image Restoration

**Questions:**

1. Fig. 4 only illustrates the feed-forward network in previous works without any modifications. It is not necessary to include those figures at least in the main paper.

2. Experiments on important image restoration tasks including image super-resolution and classic image denoising are not done.

3. See the other comments in the Weakness.